# Effects of indoor air pollution due to solid fuel combustion on physical growth of children under 5 in Sri Lanka: A descriptive cross sectional study

**Nayomi Ranathunga**[1]\*, **Priyantha Perera**[2], **Sumal Nandasena**[3], **Nalini Sathiakumar**[4], **Anuradhani Kasturiratne**[5], **Ananda Rajitha Wickremasinghe**[5]

**1** Department of Physiology, Faculty of Medicine, Wayamba University of Sri Lanka, Kuliyapitiya, Sri Lanka, **2** Department of Pediatrics, Faculty of Medicine, University of Kelaniya, Colombo, Sri Lanka, **3** Deputy Regional Director of Health Services, Office of the Regional Director of Health Services-Kalutara District, Kalutara, Sri Lanka, **4** Department of Epidemiology, School of Public Health, University of Alabama at Birmingham, Birmingham, Alabama, United States of America, **5** Department of Public Health, Faculty of Medicine, University of Kelaniya, Colombo, Sri Lanka

\* E-mail: ranayomi@gmail.com

**Data Availability Statement:** All relevant data are within the paper.

## Abstract

Solid fuel combustion is an important risk factor of morbidity. This study was conducted to determine the effect of indoor air pollution (IAP) due to solid fuel combustion on physical growth in 262 Sri Lankan children under five. Exposure was defined by the type of fuel used for cooking. Pollutant levels were measured in a subsample of households. "High" exposure group (households using biomass fuel/kerosene oil for cooking) comprised 60% of the study population; the prevalence of wasting was 19.7% and underweight was 20.4% in the entire population where 68% were from the high exposure group. Children from the "high" exposure group had significantly lower mean z-scores for weight-for-height (p = 0.047), height-for-age (p = 0.004) and weight-for-age (p = 0.001) as compared to the "low" exposure group (children of households using liquefied petroleum gas and/or electricity) after adjusting for confounders. Z-scores of weight-for-age, height-for-age and weight-for-height were negatively correlated with CO (p = 0.001, 0.018, 0.020, respectively) and $PM_{2.5}$ concentrations (p<0.001, p = 0.024 p = 0.008, respectively). IAP due to combustion of biomass fuel leads to poor physical growth.

## Introduction

Solid fuel combustion for cooking releases air borne hazardous chemicals leading to indoor air pollution (IAP). IAP is an important risk factor of the global burden of disease [1]. Although a wide range of adverse health effects have been described due to burning of unprocessed biomass (wood, animal dung, crop residues, and grasses) and coal, a significant percentage of the world's population is still using them as the main cooking fuel. Until recently IAP has been a neglected research entity. However, awareness about IAP has increased significantly recently and IAP has been identified as an important etiological agent in many disease conditions.

**Funding:** This study received financial support from the International Training and Research in Environmental and Occupational Health (ITREOH) training grant. The grant was awarded to NS and the number is (5 D43 TW05750).

**Competing interests:** The authors have declared that no competing interests exist.

Compared to any other period of life, the rate of physical growth during infancy is at its maximum. During the first year of life, weight increases by about 200% and height by about 50% [2]. After infancy, growth progresses at a slower rate until puberty, when the second peak of growth is observed. However, the rate of growth during puberty is much less compared to growth observed during infancy. Any irreversible detrimental effect on growth during early childhood will have a permanent impact on the ultimate physical growth attained by an individual.

Growth is a complex interaction between genetics and the environment. In an optimum environment, an individual's maximum growth potential is determined by genes. Nutrition is the most important environmental factor influencing physical growth. Recurrent infections, chronic diseases like poorly controlled asthma, psycho-social deprivation, congenital abnormalities, and parental substance abuse including smoking are well known risk factors of poor physical growth [3]. Though exposure to toxins is an identified risk factor for poor physical growth, little was known about indoor air pollution until recently. Results of studies carried out earlier have suggested a causal relationship between adverse physical growth and indoor air pollution [4, 5].

Unprocessed solid fuel and coal account for about 80% of the total household fuel needs in developing countries in Southeast Asia and sub-Saharan Africa [6]. Biomass fuels remain at the lower end of the energy ladder in combustion efficiency and cleanliness [7]. Hazardous air pollutants in smoke released from biomass combustion include respirable particulate matter, carbon monoxide (CO), nitrogen oxides, formaldehyde, benzene, 1–3 butadiene, polycyclic aromatic hydrocarbons (such as benzo[a]pyrene), and many other toxic organic compounds [8].

Young children are specifically affected by IAP as they spend a significant time in the kitchen with their mothers. In south India, higher prevalences of underweight and stunting at six months of age were reported among children living in households that burned biomass fuels but the association with wasting was weak [9].

Emerging evidence suggests that IAP is a major risk factor of physical growth retardation during the early years of life [10, 11]. However, most studies have concluded that further evidence is required to confirm a definite causal relationship between IAP and physical growth retardation. In this study, we investigated the effects of IAP due to solid fuel combustion on physical growth of children under five years, living in a suburban area of Sri Lanka.

## Materials and methods

### Study setting

A descriptive cross sectional study was conducted as a part of an ongoing study in which the adverse health effects of IAP on children from intrauterine life up to five years of age were studied. The study was conducted over a 2-year period from 2012 to 2014 in the Ragama Medical Officer of Health (MOH) area in the Gampaha district of Sri Lanka. The Ragama MOH area is a suburban area situated approximately 25 kilometers from the capital city of Colombo. It comprises a mixed, multi-ethnic and multi-religious community.

### Study population

Initially, 262 children under five, permanently resident in the Ragama MOH area, were selected for the study. There was a large scale study going on to assess the birth outcomes of the solid fuel combustion related air pollution. There, the pregnant females were recruited from the antenatal clinics and the air quality levels were monitored. Children living in those households where the air quality measurements were done for the main study were recruited

for this study. Children from households using biomass or kerosene as the main source of cooking fuel were considered as the 'high exposure group'. Children from households using Liquefied Petroleum Gas (LPG) or electricity for cooking comprised the 'low exposure group'.

### Inclusion and exclusion criteria

All the children whose mother was enrolled in the main study and whose parent or guardian consented for the child to participate in the study were recruited. Children with any diagnosed chronic illness, born prematurely (before 36 weeks of gestational age), with congenital abnormalities or having a recorded history of birth insults were excluded.

### Data collection

The houses of selected children were visited by a research assistant. The purpose and the nature of the study were explained to the mother/parents/guardians. An interviewer administered questionnaire was used to collect socio-demographic data and the principal source of cooking fuel. The age of the children was confirmed by checking the child health and development record, a record maintained by the routine public health services monitoring the health of the child, which contains the date of birth.

### Anthropometric measurements

Weight and height of children were measured according to standard guidelines of the World Health Organization (WHO) [12]. Length was measured in children under 2 years and height in older children to the first decimal in centimeters. Length was measured using an Infantometer with precautions taken to keep the child in the Frankfort plane and knees extended. To ensure the accuracy of measurements two examiners were involved to keep the baby in the correct position.

Standing height was measured using a stadiometer. The child was made to stand up straight against the backboard with both feet flat on the platform. Heels were kept together with toes approximately $60^0$ apart. To ensure the accuracy of measurements, the occiput, the shoulder blades, the buttocks and heels were kept in contact with the backboard, while the head was placed in the Frankfort plane.

Weight was recorded to the first decimal in kilograms. Measurements were taken with minimum clothing. A beam balance scale was used for children who could not stand. For older children a calibrated digital scale was used, as using a beam balance scale in the field for older children was not practical. Both weighing instruments were standardized against standard weights at the onset of each session. Out of 262 children initially recruited, anthropometric measurements were taken in 240 children. Some children did not cooperate to measure length/height. Hence, data analysis is based on 240 children.

### Interpretation of growth parameters

Based on international normograms, individual z-scores for each child for each anthropometric index (height-for-age, weight-for-age and weight-for-height) were obtained using WHO "Anthro" software. The mean z-scores for exposed and control group children were calculated separately for each anthropometric index. In addition, each child was categorized as either normal, stunted, wasted or underweight, based on the WHO classification [13]. A height-for-age z-score between—2 to—3 SD was categorized as moderate stunting and below -3 SD as severe stunting. A weight-for-age z-score between– 2 to -3 SD was taken as moderate underweight and less than -3 SD as severe underweight. Similarly weight-for-height z-score between -2 to -3 SD were taken as moderate wasting and severe wasting when below -3 SD [13]. Data

were entered into an EPIDATA database and analyzed using SPSS version 16 software. Anthropometric measurements were processed using WHO "Anthro" software.

## Air quality measurements

Air quality measurements were taken from a subsample of households included in the study. Air pollution levels in the kitchen during cooking of the lunch meal, the main meal of the household, were monitored using air quality measuring equipment. $PM_{2.5}$, Carbon Dioxide ($CO_2$) and Carbon Monoxide (CO) concentrations were measured using two real time monitors. While lunch was prepared, air quality measurements from the kitchen were taken for two consecutive hours with minute-to-minute recording. For accuracy of recordings, machines were calibrated and air sampling probes were mounted according to the guidelines provided by the manufacturer.

DustTrak II monitor (DUSTTRAK™) was used to measure $PM_{2.5}$ levels. TSI's Q-trak monitor was used to measure $CO_2$ and CO levels. Its sensors display real-time, simultaneous $CO_2$ and CO concentrations. The zero calibration was done before installing the instruments. The instruments were installed as given in the guidelines; in some instances, slight modifications were done to suit the available space in the kitchen. The monitor receiver inlet was kept 145 cm above the floor, 100 cm from the cooking stove and at least 150 cm away from windows and doors opening outwards. The maximum deviation from these standard specifications was less than 10cm. A standard measuring tape was used to measure distances. Recordings of air quality measurements were extracted daily and entered into the database.

## Data analysis

Data were entered into EPIDATA database and analyzed using SPSS version 20 software. Categorical data were analyzed using chi square tests, odds ratios and their 95% confidence intervals.

Z-scores of the anthropometric indices were obtained from the WHO Anthro software [14]. Anthropometric indices were compared between different categories of sociodemographic variables and exposure status using the independent sample t-test. Significant variables were then included in multiple linear regression analyses. Exposure levels were compared using Mann Whitney U statistic.

The association between z-scores of anthropometric indices and air pollution levels ($PM_{2.5}$, CO and $CO_2$) were assessed using the Pearson product moment correlation coefficient.

All analyses were done using SPSS version 20 [15].

## Ethics considerations

The nature and procedures involved in the study were explained to the parents or the guardians of eligible children. Informed written consent was obtained prior to data collection. Confidentiality of the information was ensured. Children found to have problems requiring specialized care were referred to the Colombo North Teaching Hospital at Ragama, Sri Lanka. All mothers in households using solid fuel for cooking were advised on measures to mitigate IAP. Ethics clearance was obtained from the Ethics Review Committee of the Faculty of Medicine, University of Kelaniya, Sri Lanka (P025/04/2011).

## Results

### Socio-demographic characteristics of the study population

Of the 262 children initially recruited, 155 (59%) belonged to the high exposure group, and 107 (41%) comprised the low exposure group. The socio-demographic characteristics of

**Table 1. Socio-demographic characteristics of the study population.**

| Characteristic | High exposure group[1] | Low exposure group[2] | p-value[3] |
|---|---|---|---|
| | n (%) | n (%) | |
| Sex Male | 84 (54.2) | 58 (54.2) | 0.550 |
| Female | 71 (45.8) | 49 (45.8) | |
| Age group <3 years | 83 (54.6) | 62 (60.2) | 0.225 |
| ≥3years | 69 (45.4) | 41 (39.8) | |
| Ethnicity Sinhala | 149 (96.1) | 97 (90.7) | 0.061 |
| Other | 6 (3.9) | 10 (9.3) | |
| Father's education[4] Up to O/L | 111 (72.5) | 64 (59.8) | **0.022** |
| Above O/L | 42 (27.5) | 43 (40.2) | |
| Mother's education[4] Up to O/L | 105 (68.6) | 57 (53.3) | **0.009** |
| Above O/L | 48 (31.4) | 50 (46.7) | |
| Monthly family income (SLR[5]) ≤ 20000 | 41 (26.8) | 17 (15.9) | **0.026** |
| > 20000 | 112 (73.2) | 90 (84.1) | |
| Mother employed Yes | 3 (2.0) | 3 (3.0) | 0.471 |
| No | 147 (98.0) | 97 (97.0) | |
| Having a sibling Yes | 93 (60.4) | 49 (46.2) | **0.017** |
| No | 61 (39.6) | 57 (53.8) | |
| Pre-schooling Yes | 61 (39.6) | 42 (39.6) | 0.550 |
| No | 93 (60.4) | 64 (60.4) | |
| Smoker living at home Yes | 38 (25.2) | 25 (23.6) | 0.445 |
| No | 113 (74.8) | 81 (76.4) | |

[1] High exposure group refers to children living in households using biomass as the major type of cooking fuel

[2] Low exposure group refers to children living in households using LP gas or electricity as the major type of cooking fuel

[3] based on chi square test

[4] O/L refers to General Certificate of Education Ordinary Level (11 years of formal schooling)

[5] SLR refers to Sri Lankan Rupees (1 USD≈150 SLR).

Source: adapted from Ranatunge et al. [16].

children are given in Table 1. There was no reported change in the primary cooking fuel in the study population households from the time of recruitment of the mothers to the recruitment of children. 100% of the low exposure group children's households were using liquefied petroleum gas for cooking as the primary source of fuel. Out of the high exposure group households, 94% of the households were using firewood as the primary cooking fuel.

The distribution of children in the two exposure groups were significantly different by maternal education (p = 0.009), paternal education (p = 0.022), family income (p = 0.026) and having a sibling (p = 0.017).

## Prevalence of nutritional status by study group

There were no differences in the prevalence of severe wasting, severe stunting and severe underweight between the two exposure groups. The prevalence of underweight was significantly higher among children in the high exposure group; 20.4% vs 8.2% (p = 0.007). The prevalence of stunting and wasting was higher in the high exposure group as compared to the low exposure group, but the differences were not statistically significant (Table 2).

The mean z-scores of the high exposure group were significantly lower compared to the low exposure group for all three growth parameters; weight-for-age (-1.132 vs. -0.432; p<0.001), height-for-age (-0.63 vs. 0.008; p = 0.001) and weight-for-height (-0.998 vs. -0.636;

**Table 2. Prevalence of nutritional status by study group.**

| Nutritional status | Group | | | p-value [*] |
|---|---|---|---|---|
| | Entire study population % (n = 240) | Low exposure group % (n = 98) | High exposure group % (n = 142) | |
| Wasting[1] | 17.1 (n = 41) | 13.3 (n = 13) | 19.7 (n = 28) | 0.128 |
| Severe Wasting[2] | 4.2 (n = 10) | 2.0 (n = 2) | 5.6 (n = 8) | 0.149 |
| Stunting[3] | 10.4 (n = 25) | 8.3 (n = 8) | 12.0 (n = 17) | 0.233 |
| Severe Stunting[4] | 1.3 (n = 3) | 1.0 (n = 1) | 1.4 (n = 2) | 0.637 |
| Underweight[5] | 15.4 (n = 37) | 8.2 (n = 8) | 20.4 (n = 29) | 0.007 |
| Severe Underweight[6] | 1.7 (n = 4) | 1.0 (n = 1) | 2.1 (n = 3) | 0.460 |

[*] Significance based on comparison of high and low exposure groups using chi-square test.

[1] refers to both moderate and severe wasting where the weight-for-height z-score is below -2SD

[2] refers to severe wasting where the weight-for-height z-score is below -3SD

[3] refers to both moderate and severe stunting where the height-for-age z-score is below -2SD

[4] refers to severe stunting where the height-for-age z-score is below -3SD

[5] refers to both moderate and severe underweight where the weight-for-age z-score is below -2SD

[6] refers to severe underweight where the weight-for-age z-score is below -3SD.

p = 0.032) (Table 3). Table 3 gives the associations between anthropometric indices and socio-demographic characteristics of children.

The mean weight-for-age z-scores were significantly lower in children whose parents were less educated (-0.916 vs -0.512; p = 0.013 for father's education and -0.972 vs -0.480; p = 0.002 for mother's education); the mean height-for-age z-scores were significantly lower in children whose parents were less educated compared to children of more educated parents (-0.537 vs -0.030; p = 0.011 for father's education and -0.623 vs -0.035; p = 0.001 for mother's education). Older children had significantly lower weight-for-height (-1.05 vs -0.69; p = 0.036) and weight-for-age (-0.97 vs -0.64; p = 0.032) z-scores compared to younger children. Children with siblings had a significantly lower mean z-score for weight-for-age (-0.99 vs -0.53; p = 0.003) and weight-for-height (-1.06 vs -0.60; p = 0.006) as compared to children without siblings.

As depicted in Table 4, weight-for-age, height-for-age and weight-for-height z-scores were regressed on exposure status, age, sex, monthly family income and parental education to control for potential confounding. Even after adjusting for confounders, high exposure status was a significant predictor of lower mean z-scores in all three anthropometric indices, weight-for-age (p = 0.001), height-for-age (p = 0.004) and weight-for-height (p = 0.04); height-for-age and weight-for-age mean z-scores were less by 0.5 and weight-for-height mean z-score by 0.3 in the high exposure group as compared to the low exposure group after adjusting for other variables. Children under 3 years had a significantly higher mean weight-for-age z-score of 0.338 (p = 0.037) and a mean weight-for-height z-score of 0.312 (p = 0.046) as compared to children three years or older after adjusting for other variables (Table 4). Children whose mothers were less educated had a significantly lower mean height-for-age z-score of 0.426 as compared children whose mothers were more educated after controlling for other confounding variables (p = 0.044) (Table 4).

## Air quality levels and anthropometric indices

Air quality was measured in 115 households. Details of the measurements are given in the Table 5. Carbon dioxide and carbon monoxide were measured in parts per million and particulate matter ($PM_{2.5}$) was measured in milligrams per square meter. Minute to minute data

**Table 3. Association between anthropometric indices and socio-demographic characteristics of children.**

| Characteristic | Anthropometric parameters | | | | | | | | |
|---|---|---|---|---|---|---|---|---|---|
| | Weight-for-age z-score | | | Height-for-age z-score | | | Weight-for-height z-score | | |
| | Mean | SD[1] | p-value[2] | Mean | SD | p-value | Mean | SD | p-value |
| Group | | | | | | | | | |
| High exposure[3] (n = 132) | -1.132 | 1.132 | <0.001 | -0.640 | 1.274 | 0.001 | -0.998 | 1.269 | 0.032 |
| Low exposure[4] (n = 90) | -0.435 | 1.172 | | 0.008 | 1.606 | | -0.636 | 1.291 | |
| Sex Male (n = 131) | -0.868 | 1.173 | 0.101 | -0.516 | 1.248 | 0.216 | -0.900 | 1.308 | 0.501 |
| Female (n = 109) | -0.678 | 1.189 | | -0.207 | 1.655 | | -0.788 | 1.266 | |
| Age <3 years (n = 134) | -0.642 | 1.277 | 0.032 | -0.321 | 1.600 | 0.429 | -0.692 | 1.375 | 0.036 |
| ≥3years (n = 105) | -0.971 | 1.024 | | -0.471 | 1.219 | | -1.045 | 1.148 | |
| Monthly Family Income (SLR)[5] | | | | | | | | | |
| ≤ 20,000 (n = 51) | -0.962 | 1.152 | 0.217 | -0.515 | 1.631 | 0.416 | -0.971 | 1.147 | 0.463 |
| >20,000 (n = 171) | -0.733 | 1.192 | | -0.329 | 1.400 | | -0.823 | 1.325 | |
| Father's education | | | | | | | | | |
| Up to O/L[6] (n = 148) | -0.916 | 1.124 | 0.013 | -0.537 | 1.318 | 0.011 | -0.935 | 1.254 | 0.173 |
| Above O/L (n = 74) | -0.512 | 1.264 | | -0.030 | 1.648 | | -0.693 | 1.346 | |
| Mother's education | | | | | | | | | |
| Up to O/L (n = 134) | -0.972 | 1.113 | 0.002 | -0.623 | 1.378 | 0.001 | -0.937 | 1.222 | 0.214 |
| Above O/L (n = 88) | -0.480 | 1.237 | | 0.035 | 1.483 | | -0.724 | 1.383 | |
| Having a sibling | | | | | | | | | |
| Yes (n = 131) | -0.990 | 1.142 | 0.003 | -0.490 | 1.432 | 0.181 | -1.057 | 1.245 | 0.006 |
| No (n = 109) | -0.531 | 1.186 | | -0.238 | 1.470 | | -0.601 | 1.300 | |

[1] Standard deviation

[2] based on chi square test

[3] High exposure group refers to children living in households using biomass as the major type of cooking fuel

[4] Low exposure group refers to children living in households using LP gas or electricity as the major type of cooking fuel

[5] SLR refers to Sri Lankan Rupees (1 USD≈150 SLR).

[6] O/L refers to General Certificate of Education Ordinary Level (11 years of formal schooling).

were recorded and the average value of 120 data points (2-hour continuous monitoring) were analyzed. There were significant differences in the concentrations of carbon monoxide ($p<0.001$) and the $PM_{2.5}$ ($p<0.001$) levels between the two groups; the high exposure houses had significantly higher concentrations (as much as 2–3 times more) of pollutants as compared to the low exposure group. Carbon dioxide concentrations were similar during cooking in both groups of houses.

Carbon monoxide and $PM_{2.5}$ levels were significantly negatively correlated with all three anthropometric indices (Table 6). Carbon dioxide levels were not correlated with any anthropometric index.

Exposure time was assessed and there was only one child from low exposure group and 2 children from high exposure group who were spending time near stove more than one hour per day.

## Discussion

Physical growth of a child is predetermined by genetic factors to a certain extent. However, twin studies have revealed that genetics is not the only determinant of physical growth of a child. It has been shown that monozygotic twins achieve different adult heights depending on the environments they are brought up [17]. Physical growth can be described as an outcome of

**Table 4. Summary of multiple regression analyses using growth parameters as the dependent variable.**

| Variable | Height-for-Age | | Weight-for-Height | | Weight-for-Age | |
|---|---|---|---|---|---|---|
| | Regression Coefficient | Significance (95% CI of regression coefficient) | Regression Coefficient | Significance (95% CI of regression coefficient) | Regression Coefficient | Significance (95% CI of regression coefficient) |
| Constant | 0.471 | | -0.571 | | -0.191 | |
| High exposure[1] | -0.540 | 0.004 (-0.907)–(-0.173) | -0.342 | 0.047 (-0.678)–(-0.005) | -0.510 | 0.001 (-0.808)—(-0.212) |
| Father's education (up to O/L)[2] | -0.281 | 0.189 (-0.701)– 0.139 | -0.166 | 0.396 (-0.551)– 0.219 | -0.227 | 0.191 (-0.568)–(0.114) |
| Mother's education (up to O/L)[3] | -0.426 | 0.044 (-0.841)–(-0.011) | -0.075 | 0.699 (-0.456)– 306 | -0.286 | 0.096 (-0.623)– 0.051 |
| Family income (< SLR 20000)[4] | 0.015 | 0.946 (-0.430)– 0.460 | -0.050 | 0.808 (-0.459)– 0.358 | -0.052 | 0.777 (-0.413)– 0.309 |
| Age < 3 years[5] | 0.149 | 0.417 (-0.212)– 0.510 | 0.338 | 0.046 0.007–0.669 | 0.312 | 0.037 0.019–0.605 |
| Sex (Male)[6] | -0.307 | 0.095 (-0.668)– 0.054 | -0.182 | 0.280 (-0.513)– 0.149 | -0.236 | 0.113 (-0.529)– 0.057 |

[1]Reference group is low exposure group using LPG and electricity for cooking.

[2]Reference group is father's education above ordinary level

[3]Reference group is mother's education above ordinary level

[4] Reference group is having monthly family income ≥Sri Lanka rupees (SLR) 20,000 (1 USD≈150 SLR)

[5]Reference group is children aged ≥3 years

[6]Reference group is female children.

a complex interaction between genetic and non-genetic factors. Important non-genetic factors influencing physical growth are nutrition, infections/diseases, psycho-social well-being, physical activity and environmental factors [17].

IAP has been increasingly implicated as a preventable cause of morbidity and mortality among children. IAP is also considered an important environmental risk factor for poor physical growth in children [18]. Children under five stay indoors most of the time and are likely to stay near the stove, while the mother prepares meals. Hence, children under five are more likely to be exposed to hazards of indoor air pollution, resulting from combustion of biomass fuel. High minute ventilation, immature immune systems and vigorous physical activities make children more vulnerable to adverse health effects of IAP than adults.

There are only a very few studies that have addressed effects of IAP on physical growth. Weight-for-age and weight-for-height are affected by acute changes in a child's nutrition and environment, while height-for-age indicates chronic effects [19]. According to our study, all three anthropometric indices (weight-for-age, height-for-age and weight-for-height) were significantly affected by IAP. Even when adjustments were made for confounding factors, IAP was a significant predictor of poor physical growth in children under five. Our results indicate that IAP has both acute and chronic effects on physical growth of children.

**Table 5. Air quality measurements in selected houses.**

| Exposure | | Number of households | Median | Interquartile range | Significance |
|---|---|---|---|---|---|
| CO | High exposure | 64 | 1.90 ppm | 1.20 ppm -3.57 ppm | <0.001 |
| | Low exposure | 51 | 1.20 ppm | 0.85 ppm– 1.5 ppm | |
| $PM_{2.5}$ | High exposure | 65 | 0.58 mg/m$^3$ | 0.17 mg/m$^3$–1.71 mg/m$^3$ | 0.881 |
| | Low exposure | 56 | 0.15 mg/m$^3$ | 0.06mg/m$^3$–0.27 mg/m$^3$ | |
| $CO_2$ | High exposure | 66 | 547.75 ppm | 449.0 ppm– 647.25 ppm | <0.001 |
| | Low exposure | 52 | 538.5 ppm | 454.0 ppm– 634.63 ppm | |

**Table 6.  Correlation between air quality levels and anthropometric indices.**

| | | z-scores of anthropometric measurements | | |
| --- | --- | --- | --- | --- |
| | | weight-for-age | height-for-age | weight-for-height |
| Carbon Monoxide | Pearson Correlation Coefficient | -0.354 | -0.251 | -0.245 |
| | Significance | 0.001 | 0.018 | 0.020 |
| | N | 89 | 89 | 89 |
| Carbon Dioxide | Pearson Correlation Coefficient | 0.036 | -0.026 | 0.050 |
| | Significance | 0.730 | 0.802 | 0.631 |
| | N | 94 | 94 | 94 |
| Particulate Matter 2.5 | Pearson Correlation Coefficient | -0.356 | -0.233 | -0.272 |
| | Significance | <0.001 | 0.024 | 0.008 |
| | N | 94 | 94 | 94 |

Moderate underweight was significantly higher among children in the high exposure group. Moderate stunting was also higher in the high exposure group compared to the low exposure group, but not statistically significant. The rate of weight increase during childhood is several folds faster than length/height [2]. Therefore, an adverse influence on growth will be evident early in weight than in length/height. That is the basis of monitoring weight monthly and length once in three months during first two years. This probably explains the finding of our study, where stunting was not associated with IAP; the findings of a study amongst older children may be different as stunting takes a longer time to manifest with chronic exposure. A study from Bangladesh also reported similar observations as the early life prevalence of under-weight is higher than stunting [20].

Although z-scores of anthropometric indices were associated with exposure status after adjusting for other variables, IAP was not directly associated with the prevalence of severe forms of wasting, stunting or underweight. This suggests that effects of IAP on growth may not be as strong as malnutrition or recurrent infections. However, as households with higher IAP are likely to have other adverse conditions for growth like poor parental education, lower income, malnutrition, infections, and worm infestations, IAP would contribute significantly to the cumulative effects on physical growth. In the high exposure group, weight-for-height z-scores were significantly less in older children compared to younger children. This suggests that the effects of IAP may accumulate over the years.

CO and $PM_{2.5}$ are two well-known pollutants present in smoke from biomass combustion [21]. Combustion of biomass in an open stove without a chimney emits particulate matter (PM) concentrations between 2000 to 15 000 $\mu g/m^3$, a level many times higher than the worst outdoor settings [22, 23]. Release of these pollutants is minimal with LP gas and electricity. In our study, all three anthropometric indices were negatively correlated with CO and $PM_{2.5}$ levels.

The exact mechanism of how IAP adversely affects physical growth is not explained. Smoke released from biomass combustion contains many hazardous substances such as respirable particulate matter, CO, nitrogen oxides, formaldehyde, benzene, 1–3 butadiene, and polycyclic aromatic hydrocarbons (such as benzo[a]pyrene). These can exert a direct toxic effect on the growth plates and growing tissues or may cause chronic ill health in children resulting in growth retardation. Chronic respiratory conditions like asthma which have a direct adverse effect on growth have been shown to increase with IAP [24, 25].

Parental education, family income, having siblings and gender can influence physical growth. Even when effects of these confounding factors were eliminated through regression analysis the effects of IAP on physical growth remained significant. Therefore, in lower socio-

economic conditions, IAP may act along with other risk factors to hamper physical growth of children. If it is not possible to provide efficient fuels like LP gas and electricity to all households, at least attempts should be made to improve the cook stoves to minimize IAP. In addition, mothers should be educated to keep children away from stoves during cooking.

## Limitations

We were able to measure air quality levels in only a subsample of households due to limited resources which is a limitation of our study. Further, we were unable to measure outdoor air pollution levels that may have had an effect on indoor air pollution levels. As the study population was under 5 children who generally stay indoors most of the time during the day, we assumed that exposure to outdoor air pollution will have a minimal effect. All children were from the same geographic area and would likely have been exposed to the same levels of outdoor pollution.

We measured air pollution levels over a two-hour period during the preparation of the lunch meal, the main meal that is cooked in most households. Based on the construction of houses and air circulation within houses it is possible that air pollution levels may have been higher in areas other than in the kitchen where children may have been and beyond the time after we stopped monitoring pollutant levels. We acknowledge this as a limitation and it is likely that it may have affected our results. We were only able to measure CO, $CO_2$ and $PM_{2.5}$; this limited our ability to assess interactions with other pollutants.

## Conclusion

Biomass combustion results in significant IAP compared to LP gas or electricity. IAP was significantly negatively associated with physical growth of children. Parents should be advised to keep children away from the stove while cooking.

## Author Contributions

**Conceptualization:** Nayomi Ranathunga, Sumal Nandasena, Ananda Rajitha Wickremasinghe.

**Data curation:** Nayomi Ranathunga, Sumal Nandasena.

**Formal analysis:** Nayomi Ranathunga.

**Funding acquisition:** Nalini Sathiakumar.

**Investigation:** Nayomi Ranathunga, Nalini Sathiakumar.

**Methodology:** Priyantha Perera, Sumal Nandasena, Ananda Rajitha Wickremasinghe.

**Project administration:** Nayomi Ranathunga, Priyantha Perera, Sumal Nandasena, Nalini Sathiakumar, Ananda Rajitha Wickremasinghe.

**Resources:** Priyantha Perera, Ananda Rajitha Wickremasinghe.

**Supervision:** Priyantha Perera, Sumal Nandasena, Nalini Sathiakumar, Anuradhani Kasturiratne, Ananda Rajitha Wickremasinghe.

**Validation:** Ananda Rajitha Wickremasinghe.

**Writing – original draft:** Nayomi Ranathunga.

**Writing – review & editing:** Priyantha Perera, Sumal Nandasena, Nalini Sathiakumar, Anuradhani Kasturiratne, Ananda Rajitha Wickremasinghe.

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
