## [Decision Letter · Decision Letter 0]

29 Dec 2020

PONE-D-20-37412

Effects of indoor air pollution due to solid fuel combustion on physical growth of children under 5

PLOS ONE

Dear Dr. Ranathunga,

Thank you for submitting your manuscript to PLOS ONE. After careful consideration, we feel that it has merit but does not fully meet PLOS ONE’s publication criteria as it currently stands. Therefore, we invite you to submit a revised version of the manuscript that addresses the points raised during the review process.

Please pay special attention to the reviewers' comments and suggestions about some details of the methods and data analysis, along with study design and clarification of cofounding factors.

We look forward to receiving your revised manuscript.

Kind regards,

Qinghua Sun, MD, PhD

Academic Editor

PLOS ONE

Journal Requirements:

3. We noted in your submission details that a portion of your manuscript may have been presented or published elsewhere.

"Data in Table 1 is the same as some of the data in Table 1 of Ranatunge at al (give reference). These data relate to sociodemographic data which are the same as the two publications are from the same study but the objectives and focus of the two manuscripts are different. Therefore the same baseline characteristics are common to the same study population. The published manuscript has been cited as a footnote in this submission."

Please clarify whether this publication was peer-reviewed and formally published. If this work was previously peer-reviewed and published, in the cover letter please provide the reason that this work does not constitute dual publication and should be included in the current manuscript.

Reviewers' comments:

Reviewer's Responses to Questions

**Comments to the Author**

1. Is the manuscript technically sound, and do the data support the conclusions?

Reviewer #1: Yes

Reviewer #2: Yes

2. Has the statistical analysis been performed appropriately and rigorously? 

Reviewer #1: Yes

Reviewer #2: I Don't Know

3. Have the authors made all data underlying the findings in their manuscript fully available?

Reviewer #1: Yes

Reviewer #2: No

4. Is the manuscript presented in an intelligible fashion and written in standard English?

Reviewer #1: Yes

Reviewer #2: Yes

5. Review Comments to the Author

Reviewer #1: Dec-10-2020

Comments to the manuscript PONE-D-20-37412:

In the present manuscript, the authors used a population-based descriptive cross sectional study to address the association between solid fuel combustion-derived indoor air pollution and the physical growth in children less than five years old in Ragama MOH, Sri Lanka. The exposure was defined by cooking fuel types and pollutant levels were measured in a household subsample. They reported that the high-exposure children had significantly lower mean z-scores for weight-for-height, height-for-age and weight-for-age, and the scores were negatively correlated with indoor CO and PM2.5 concentrations related to cooking. They concluded that indoor air pollution by biomass fuel combustion leads to poor physical growth. The study is of significance; however, there are methodological concerns about the research as presented.

1. Indoor air pollution is strongly spatiotemporally dependent. A child may be hurt more severely by indoor air pollution as the cooking lasts longer even if the pollution is comparable or lighter. Analysis with the absence of exposure time will, therefore, compromise the results.

2. In the high-exposure group, unprocessed biomass, which includes wood, grasses, crop residues or animal dung, was used for cooking. Coal and kerosene were also used. In the low-exposure group, they used liquefied petroleum gas or electricity, with the former producing particles, SO2, NOx, CO, non-methane total hydrocarbons (incomplete combustion), etc., at a lesser extent. Since different fuel combustion produces different air pollutants, which may exert different harmful effects on health, the authors need to present in detail the situation of households using different fuels. Besides, they should pay attention to the history of fuels used, because the type of fuels might be changed during the study period.

3. Air quality was measured in 115 out of 242 households. Why were the measurements not taken from all households? Similarly, analysis was performed based on data from the subgroups (e.g., Table 7). Comparisons conducted with smaller sample sizes would potentially bias the results.

4. Inclusion and exclusion criteria were not presented. Was the child with premature birth, obesity or other abnormal conditions (prenatally and postnatally) included? Are there any twins?

5. Description for data analysis is not sufficient. How to compare the anthropometric parameters, e.g., values of weight-for-age z-scores expressed as mean and SD (Table 03), between the two groups? There is a lack of description regarding confounding factors in Method session. The authors should give the reason why they chose exposure status, age, sex, monthly family income and parental education as the confounders. Multicollinearity should be considered for tightly correlated variables if any. Did they consider the co-effects of outdoor-level air pollution, which may not be negligible especially in developing countries?

6. There are strong relationships of different toxic pollutants to one another with regard to air pollution on health. There are no models of multi-pollutants to reflect the texture of air pollutants. The average concentrations of CO, CO2 and PM2.5 for both groups should be given. What did the authors evaluate the temperature and humidity for?

7. The authors may have to evaluate the weakness of the study in the Discussion session.

8. Other points:

(1) “Any detrimental effect on growth during early childhood is irreversible” (Line 51): This may not be true because some effects are temporary and reversible.

(2) form → from (Line 92).

(3) ‘exposed group’ (Line 96) and “control group (Line 97): It is more appropriate to use “high exposure group” and “low exposure group”, as used by the authors in other places.

(4) Statements such as “… to collect socio-demographic data and the principal source of cooking” (Line 102) should go to the Method session.

(5) “PM2.5 and Carbon Monoxide (CO) concentrations were measured …” (Line 144-145): It should be PM2.5, CO2 and CO to be measured.

(6) I suggest the sublevel headings be used for the Results session.

(7) “Stunting and wasting were …” (Line 200-201)：→ The prevalence (or ratio) of stunting and wasting was…

(8) “P” whether in uppercase (as in Table 3) or in lowercase (as in Table 2) should be consistent.

(9): Table 1 (Page 9): “Family income” : Is it monthly income ? (indicated in Line 242)

(10): “Entire study population” column in Table 2 (Page 10): “n = 1.7” ?

(11): Line 213 and 214: “vs” → “vs.”.

(12): Line 235: “… to children three …”→ “… to children of three …”; “(Tables 4 and 6)” → (Tables 4-6).

(13): Line 306-308: Lacking reference(s).

(14): “the findings of a study amongst older children may be different”: No supportive data.

(15): “However, as households with higher IAP … , IAP would contribute … on physical growth” (Line 327-330): The authors give insufficient rationales that IAP contributed significantly because IAP is not definitely correlated with other adverse conditions.

(16): Line 336: “mg/m3” → μg/m3

(17): “respirable particulate matter” (Line 341): Do they refer to “inhalable particulate matter”? If so, “PM2.5” that follows can be omitted.

(18): I suggest the name of country be added in the title and deleted in “Key Words”.

(19): Line 371-373: Since it is not necessary to repeat the funding, the Acknowledgements session can be deleted.

Reviewer #2: This is an important study on the health impact of indoor air pollution on child growth. On the whole, the study is designed appropriately, the data collected with adequate care, and the manuscript well written. My main comments concern the choices that the authors have made in their analysis, presentation, and write-up.

1. As this is an observational study, the authors may wish to change the title to reflect that appropriately.

2. The supplementary information is another article on the effect of indoor air pollution on childhood respiratory diseases published by the authors. Is this intentional or a mistake?

3. The choice of covariates in the regression is not clear to me. Why did the authors not use all the covariates in the regression analysis?

4. The results in tables 4-6 can probably be presented more succinctly (in one table).

5. Did the authors run regressions with the quantitative measures of air pollutants as the independent variable(s) rather than exposure group? I would be interested in seeing the results for these regressions as well.

6. The manuscript does not interpret the regression estimates comprehensively (sign, significance and size). The size of the effect should probably be described in the results section.

7. In the discussion, the authors refer to the results of the correlation between severe anthropometric failure and exposure group after discussing the results of the regressions. If the authors wish to establish the (absence of a) relationship between severe anthropometric failure and exposure group, they might want to run regressions with severe stunting, severe wasting, and severe underweight as the dependent variable.

8. The limitations of the study should be described in the discussion.

6. PLOS authors have the option to publish the peer review history of their article (what does this mean?). If published, this will include your full peer review and any attached files.

Reviewer #1: No

Reviewer #2: No

---

## [Author Response · Author response to Decision Letter 0]

10 Apr 2021

Responses to reviewer comments

Reviewer comment – Reviewer 01 

Comment 01

Indoor air pollution is strongly spatiotemporally dependent. A child may be hurt more severely by indoor air pollution as the cooking lasts longer even if the pollution is comparable or lighter. Analysis with the absence of exposure time will, therefore, compromise the results. 

Response to comment 01

We agree with the reviewer’s comment. Exposure time was assessed and there was only one child from the low exposure group and 2 children from high exposure group who spent time near stove more than one hour during the exposure measurement. Furthered the maximum time spent near the stove per day by a child was 2 hours (one child)(revised in the manuscript line 282-283).

As suggested by the reviewer, indoor air pollution is spatiotemporally dependent. We were only able to measure exposure for a two hour period during the cooking of lunch, the main meal of the family. We have mentioned this as a limitation (line numbers 354 to 358 in revised manuscript). 

Comment 02

In the high-exposure group, unprocessed biomass, which includes wood, grasses, crop residues or animal dung, was used for cooking. Coal and kerosene were also used. In the low-exposure group, they used liquefied petroleum gas or electricity, with the former producing particles, SO2, NOx, CO, non-methane total hydrocarbons (incomplete combustion), etc., at a lesser extent. Since different fuel combustion produces different air pollutants, which may exert different harmful effects on health, the authors need to present in detail the situation of households using different fuels. Besides, they should pay attention to the history of fuels used, because the type of fuels might be changed during the study period. 

Response to comment 02

We monitored the use of different fuels during the study. The mothers of these children were recruited during their pregnancy and continuously followed. Since, recruitment of these mothers there was no change in the principal cooking fuel. 100% of the low exposure group households were using liquid petroleum gas for cooking as the primary source of fuel. In the high exposure group households, 94% of the households were using firewood as the primary cooking fuel. As we present baseline results of the study on children under 5, any change in fuel is unlikely to have affected the results.(Revised in the manuscript line 191-195)

Comment 03

Air quality was measured in 115 out of 242 households. Why were the measurements not taken from all households? Similarly, analysis was performed based on data from the subgroups (e.g., Table 7). Comparisons conducted with smaller sample sizes would potentially bias the results. 

Response to comment 03

Due to the limited availability of resources, we measured the air quality levels in a subsample of the cohort. That was a limitation of the research and included in the limitation section. (Revised in the manuscript line 353-355)

Comment 04

Inclusion and exclusion criteria were not presented. Was the child with premature birth, obesity or other abnormal conditions (prenatally and postnatally) included? Are there any twins? 

Response to comment 04

Inclusion and exclusion criteria have been included.(Revised in the manuscript line 99-103) 

Comment 05

Description for data analysis is not sufficient. How to compare the anthropometric parameters, e.g., values of weight-for-age z-scores expressed as mean and SD (Table 03), between the two groups? There is a lack of description regarding confounding factors in Method session. The authors should give the reason why they chose exposure status, age, sex, monthly family income and parental education as the confounders. Multicollinearity should be considered for tightly correlated variables if any. Did they consider the co-effects of outdoor-level air pollution, which may not be negligible especially in developing countries? 

Response to comment 05

Z scores for anthropometric measurements were obtained from WHO Anthro software. Mean and SD was calculated for those values. Comparisons between the high and low exposure groups was done using independent sample t-tests. The anthropometric parameters for different socio-demographic groups were also compared using the independent sample t-test. 

The reason why age, sex, monthly family income and parental education were considered were included as potential confounders as they were significant on bivariate analysis (unadjusted). In the multiple regression model, multicollinearity between independent variables would have been considered in the final output. 

We did not consider co-effects of outdoor air pollution as we did not measure. We have included it as a limitation. (Revised in the manuscript line 354-355)

Comment 06

There are strong relationships of different toxic pollutants to one another with regard to air pollution on health. There are no models of multi-pollutants to reflect the texture of air pollutants. The average concentrations of CO, CO2 and PM2.5 for both groups should be given. What did the authors evaluate the temperature and humidity for? 

Response to comment 06

We were able to measure only CO, CO2 and PM2.5. We have included a statement as a limitation (Revised in the manuscript line 364-365) The air quality measurements were obtained with the temperature and humidity being automatically measured by the testing instrument. As we did not observe any significant variation in those measurements, we did not analyze those data. Therefore, that part was not included in the manuscript. Data presented in the table 06 are related to the sub sample of households (115) where 55% belonged to high exposure group. An additional table has been included (Table 05)

Comment 07

The authors may have to evaluate the weakness of the study in the Discussion session. 

Response to comment 07

Included in the revised manuscript giving the limitations as a subsection 

Comment 08

“Any detrimental effect on growth during early childhood is irreversible” (Line 51): This may not be true because some effects are temporary and reversible. 

Response to comment 08

The meaning of “detrimental effect” defined here is about irreversible effects and the sentence is revised in the manuscript

Comment 09

form → from (Line 92). 

Response to comment 09

Revised the error

Comment 10

‘exposed group’ (Line 96) and “control group (Line 97): It is more appropriate to use “high exposure group” and “low exposure group”, as used by the authors in other places. 

Response to comment 10

Corrected in the manuscript

Comment 12

Statements such as “… to collect socio-demographic data and the principal source of cooking” (Line 102) should go to the Method session. 

Response to comment 12

It is in the methods section

Comment 13

“PM2.5 and Carbon Monoxide (CO) concentrations were measured …” (Line 144-145): It should be PM2.5, CO2 and CO to be measured. 

Response to comment 13

This has been corrected in the revised manuscript.

Comment 14

I suggest the sublevel headings be used for the Results session. 

Response to comment 14

Included in the manuscript

Comment 15

“Stunting and wasting were …” (Line 200-201) → The prevalence (or ratio) of stunting and wasting was.

Response to comment 15

Corrected in the manuscript 

Comment 16

“P” whether in uppercase (as in Table 3) or in lowercase (as in Table 2) should be consistent. 

Response to comment 16

Converted to lowercase in table 3.

Comment 17

Table 1 (Page 9): “Family income”: Is it monthly income? (indicated in Line 242) 

Response to comment 17

Corrected in tables 1 and 3.

Comment 18

“Entire study population” column in Table 2 (Page 10): “n = 1.7”? 

Response to comment 18

It was a typing error. Corrected in the revised manuscript 

Comment 19

Line 213 and 214: “vs” → “vs.”. 

Response to comment 19

Corrected in the manuscript

Comment 20

Line 235: “… to children three …”→ “… to children of three …”; “(Tables 4 and 6)” → (Tables 4-6). 

Response to comment 20

Corrected in the manuscript

Comment 21

Line 306-308: Lacking reference(s). 

Response to comment 21

Reference given 

Comment 22

“the findings of a study amongst older children may be different”: No supportive data. 

Response to comment 22

It was mentioned in the discussion where it needs a follow up for a longer time and that has not been assessed in this study. (Revised in the manuscript line 317)

Comment 23

“However, as households with higher IAP … , IAP would contribute … on physical growth” (Line 327-330): The authors give insufficient rationales that IAP contributed significantly because IAP is not definitely correlated with other adverse conditions. 

Response to comment 23

In this section, it was meant to address the multifactorial effects of growth which was addressed in this study and not completely addressed in some aspects such as nutrition. Therefore, we discussed the cumulative effect which can be a reinforcing factor for growth faltering.

Comment 24

Line 336: “mg/m3” → μg/m3 

Response to comment 24

Corrected in the manuscript

Comment 25

“respirable particulate matter” (Line 341): Do they refer to “inhalable particulate matter”? If so, “PM2.5” that follows can be omitted. 

Response to comment 25

Corrected in the manuscript

Comment 26

I suggest the name of country be added in the title and deleted in “Key Words”. 

Response to comment 26

Corrected in the manuscript

Comment 27

Line 371-373: Since it is not necessary to repeat the funding, the Acknowledgements session can be deleted. 

Response to comment 27

Corrected in the manuscript

Responses to reviewer comment – Reviewer 02 

Comment 01

As this is an observational study, the authors may wish to change the title to reflect that appropriately. 

Response to comment 01

Corrected in the manuscript

Comment 02

The supplementary information is another article on the effect of indoor air pollution on childhood respiratory diseases published by the authors. Is this intentional or a mistake? 

Response to comment 02

It was given intentionally as this is the same study population and the same sociodemographic data are repeated in both.

Comment 03

The choice of covariates in the regression is not clear to me. Why did the authors not use all the covariates in the regression analysis? 

Response to comment 03

We have now explained it under data analysis. Only the significant covariates on bivariate analyses were included in the multiple linear regression analyses.(Revised in the manuscript line 170-173)

Comment 04

The results in tables 4-6 can probably be presented more succinctly (in one table). 

Response to comment 04

Corrected in the manuscript

Comment 05

Did the authors run regressions with the quantitative measures of air pollutants as the independent variable(s) rather than exposure group? I would be interested in seeing the results for these regressions as well. 

Response to comment 05

We did not do this analysis as air pollutant levels were measured in only 114 households.

Comment 06

The manuscript does not interpret the regression estimates comprehensively (sign, significance and size). The size of the effect should probably be described in the results section. 

Response to comment 06

This has been done in the revised manuscript.

Comment 07

In the discussion, the authors refer to the results of the correlation between severe anthropometric failure and exposure group after discussing the results of the regressions. If the authors wish to establish the (absence of a) relationship between severe anthropometric failure and exposure group, they might want to run regressions with severe stunting, severe wasting, and severe underweight as the dependent variable. 

Response to comment 07

We did not do this analysis as the numbers with severe forms of malnutrition were too few to perform any analysis.

Comment 08

The limitations of the study should be described in the discussion. 

Response to comment 08

Added as a separate section. (Revised in the manuscript line 353-365)

---

## [Decision Letter · Decision Letter 1]

12 May 2021

Effects of indoor air pollution due to solid fuel combustion on physical growth of children under 5 in Sri Lanka: A descriptive cross sectional study

PONE-D-20-37412R1

Dear Dr. Ranathunga,

We’re pleased to inform you that your manuscript has been judged scientifically suitable for publication and will be formally accepted for publication once it meets all outstanding technical requirements.

Kind regards,

Qinghua Sun, MD, PhD

Academic Editor

PLOS ONE

Additional Editor Comments (optional):

Reviewers' comments:

Reviewer's Responses to Questions

**Comments to the Author**

1. If the authors have adequately addressed your comments raised in a previous round of review and you feel that this manuscript is now acceptable for publication, you may indicate that here to bypass the “Comments to the Author” section, enter your conflict of interest statement in the “Confidential to Editor” section, and submit your "Accept" recommendation.

Reviewer #1: All comments have been addressed

2. Is the manuscript technically sound, and do the data support the conclusions?

Reviewer #1: Yes

3. Has the statistical analysis been performed appropriately and rigorously? 

Reviewer #1: Yes

4. Have the authors made all data underlying the findings in their manuscript fully available?

Reviewer #1: Yes

5. Is the manuscript presented in an intelligible fashion and written in standard English?

Reviewer #1: Yes

6. Review Comments to the Author

Reviewer #1: Comments to the revised version of the manuscript (No. PONE-D-20-37412R1):

The authors have answered all the questions well and revised what necessary in the text, including the addition of “Limitations” statement. With the addition of exclusion criteria, i.e., “Children with any diagnosed chronic illness, born prematurely (before 36 weeks of gestational age), with congenital abnormalities or having a recorded history of birth insults were excluded” (P.99-102), it is anticipated that there might be a few children who met the criteria to be excluded. I wonder why none of the children was ruled out. The authors may need to check the data carefully.

Some minor points:

1. “Data were entered … using SPSS version 16 software” (L.159 in “Data Analysis”) and “All analyses were done using SPSS version 20” (L.168): Which version is right?

2. “databaseand” → database and (L.158)

“software.Categorical” → software. Categorical (L.159)

“Ethicsconsiderations” → Ethics considerations (L.169)

“wereusing” (L.185) → were using

“whogenerally” (L.354) → who generally

7. PLOS authors have the option to publish the peer review history of their article (what does this mean?). If published, this will include your full peer review and any attached files.

Reviewer #1: No

---

## [Editor Report · Acceptance letter]

17 May 2021

PONE-D-20-37412R1 

Effects of indoor air pollution due to solid fuel combustion on physical growth of children under 5 in Sri Lanka : A descriptive cross sectional study 

Dear Dr. Ranathunga:

I'm pleased to inform you that your manuscript has been deemed suitable for publication in PLOS ONE. Congratulations! Your manuscript is now with our production department. 

Kind regards, 

on behalf of

Dr Qinghua Sun 

Academic Editor

PLOS ONE